# A Review on Electronic Health Record Text-Mining for Biomedical Name Entity Recognition in Healthcare Domain

**DOI:** 10.3390/healthcare11091268

**Published:** 2023-04-28

**Authors:** Pir Noman Ahmad, Adnan Muhammad Shah, KangYoon Lee

**Affiliations:** 1School of Computer Science, Harbin Institute of Technology, Harbin 150001, China; 2Department of Computer Engineering, Gachon University, Seongnam 13120, Republic of Korea

**Keywords:** healthcare, biomedical, data-mining, bNER, electronic health records

## Abstract

Biomedical-named entity recognition (bNER) is critical in biomedical informatics. It identifies biomedical entities with special meanings, such as people, places, and organizations, as predefined semantic types in electronic health records (EHR). bNER is essential for discovering novel knowledge using computational methods and Information Technology. Early bNER systems were configured manually to include domain-specific features and rules. However, these systems were limited in handling the complexity of the biomedical text. Recent advances in deep learning (DL) have led to the development of more powerful bNER systems. DL-based bNER systems can learn the patterns of biomedical text automatically, making them more robust and efficient than traditional rule-based systems. This paper reviews the healthcare domain of bNER, using DL techniques and artificial intelligence in clinical records, for mining treatment prediction. bNER-based tools are categorized systematically and represent the distribution of input, context, and tag (encoder/decoder). Furthermore, to create a labeled dataset for our machine learning sentiment analyzer to analyze the sentiment of a set of tweets, we used a manual coding approach and the multi-task learning method to bias the training signals with domain knowledge inductively. To conclude, we discuss the challenges facing bNER systems and future directions in the healthcare field.

## 1. Introduction

### 1.1. Background of Biomedical Name Entity Recognition (bNER)

Information extraction in biomedical informatics is essential for handling medical literature, which grows exponentially and extracts information in medical research [1]. Biomedical informatics analyze and identify biomedical name entity recognition (bNER) and healthcare as biomedical entities in electronic health records (EHRs), such as drugs (medicines), proteins from compound genes, and diseases in unstructured medical texts [2,3,4,5,6]. With the unprecedented expansion of electronic medical records (EMR) worldwide, millions of pieces of data have been collected since the publication of the basic norms of EMR [7,8,9,10]. The correlated features, temporal dynamics, and uncertainty propose a variational recurrent imputation network that combines imputation and the prediction of treatment information. Herbal treatment in Chinese medicine has evolved, over thousands of years, into a unique and complicated system [11,12]. Western countries are getting more into it as an alternative and complementary medicine. Unfortunately, this unstructured data cannot be directly used in the clinical practice setting to improve patient care [13]. Based on 4 years of EHR (2018–2022), covering the entire region or country, an accurate and transparent Bayesian model is constructed on knowledge of how to test the model [14]. Data-driven studies, health outcomes management, and clinical decision-making are increasingly made possible with the improvement of Information Technology’s positive impact [15].

In addition, name entity recognition (NER) aims to identify text belonging to predefined semantic types such as a person, location, organization (P-L-O) [16], etc. NER is an information extraction tool and plays a variety of roles in natural language processing (NLP), question answering (Q/A), knowledge-based (KB) [17] automatic text summarization (ATS), sentiment analysis [18], machine translation (MT) [19], and information retrieval (IR) [20]. The named entity (NE) is identified by the names P-L-O (medical code, amounts, period, and percentage expression) into predefined classes in the EHR [21,22]. Recently, NER has gained interest and is being discussed at several scientific conferences (e.g., CoNLL03 [23], SemEval [24], and IREX). NE describes something or someone such as a place, person, or location [25]. Semantic (sense) ambiguity, as well as the model’s stability and generalizability, is one of the most common problems with NER. Moreover, proper nouns may have multiple meanings depending on the context. Mansouri et al. [26] limited natural kind terms and proper names such as biological species and substances. However, Kripke et al. [27] confined the word-named entities to a rigid designator. NEs are domain-specific (proteins, enzymes, and genes) and general (person/location). Specifically, this study examines generic and domain-specific bNER in unilingual (English language) healthcare text-mining propaganda (https://corenlp.run/ (accessed on 03 January 2023)). Furthermore, this study is beyond the scope of NER’s work on multilingualism.

Moreover, the development of appropriate models in NLP has made it possible for recent advances in propaganda detection to be achieved. Social media, fake news, and various propaganda techniques have contributed to increased misinformation on the Internet in the last ten years [28,29,30,31]. In addition to threatening the social order, broadcasting, economy, climate change, health, and democracy, these techniques are linked to terrorism [32,33]. An individual or group may commit a propaganda act when they deliberately intend to influence the actions or opinions of other individuals or groups in a way that will lead them toward achieving a specific goal [34]. According to O’Donnell et al. [35], propaganda is a deliberate and systematic process that influences a person’s opinions, thoughts, and behaviors to achieve the campaign’s intent. It is important to distinguish propaganda from lies, distortions, and deception, where distorted messages are aimed at misleading or deceiving the public, either intentionally or unintentionally [36,37]. The concept of propaganda, at the level of social media, should be well understood since it affects the harmony of the social network. The short-text propaganda stream classification is extended using a large-scale semantic network developed from a Web corpus derived from a large collection of short-text propaganda streams [38]. Currently, a significant challenge is facing one of the most challenging and significant tasks in EHR: the classification of short-text streams.

### 1.2. Impact of Biomedical EHRs Analysis

In biomedical analysis, NLP is becoming more common, which helps process and analyze EHRs automatically. The pre-trained model enables representations, a method of computing the contributions of different diseases and admissions, to maximize EHR data utilization and fully utilize healthcare domain information [39]. For cross-domain medical applications, a missingness-aware feature extraction network is an additional feature for medical analysis by incorporating healthcare conditions into the feature extraction [40]. EHR mining integration methods are applied to real-time practice parameters using meta-learning methods that automatically adjust sample weights based on real-time parameters [41]. Random noise in the samples, as well as sample bias, may lead to issues with overfitting and poor generalization. NE corpora that humans annotate are essential for training and testing automated bNER systems. Unified clinical terminology systems (UCTS), clinical oncology, and medical databases, such as DrugBank, contribute to bNER in clinical records in English, including the unified medical language system and others [42]. Certain entity types in the EHR are annotated, including drugs, treatments, symptoms, tests, temporary words, anatomy, and operations [43,44,45,46,47]. In COVID-19, more organizations use healthcare propaganda to spread awareness and promote prevention and control. Organizations responsible for public health should monitor social media communication to discover what rumors and myths are circulating or actively promoted in the community, what misconceptions exist about various treatments and diseases, and what ideas, problems, and rumors concern the public the most. This study reviews healthcare entries based on clinical records used to treat various diseases.

### 1.3. Motivations of the Study

This study presents an overview of the shift in techniques from machine learning to handcrafted rules. Kundeti et al. [48] focused on NER from the perspective of the clinical domain in 2016. Particularly, we focus on the fallacies, challenges, and analyses of the NE approach to clinical reports. The recent review mentions new domains bNER [49] and clinical NER (cNER) [50]. Accordingly, the existing review focuses on feature-based ML models rather than DL-based and bNER systems in healthcare. However, deep-learning (DL) techniques are not included in their analysis. DL has been applied to bNER over the last few years, significantly advancing the state-of-the-art performance for tagging chunking, name entities, and semantic labels [51,52,53]. As a result of this trend, this review is focused on mining text from electronic health records to recognize the names of biomedical entities within the healthcare domain.

This paper discusses techniques that influence bNER in EHR corpora, addressing issues of both approaches, which improve the annotation process and quality. This study aims to identify factors influencing bNER performance, healthcare issues, and associated challenges by comparing their DL architectures. Although bNER studies have thrived for a few decades now, the field of reviews on bNER studies has developed more, and to our knowledge, few reviews in this field have been published to date. The following are our main contributions:This study examines DL techniques in bNER, as well as their applications in propagandist text in electronic health records, and it provides researchers and practitioners with the current understanding of these techniques. A major focus of this study is to consolidate bNER and provide the research community with relevant corpora for bNER.This study presents a broad review of DL techniques for bNER entries in clinical records, for online or electronic treatment, for the diagnosis and treatment of various diseases.The study proposes a new classification that categorizes DL-based bNER approaches, input representations, and encoders and decoders. The proposed classification receives context for tag decoding and predicts labels for words based on given sequences.Additionally, this study examines the most representative DL methods for solving bNER problems and challenges. As a result, we summarize the challenges and future directions facing healthcare propaganda and bNER systems in the coming years.Information extraction (IE) and NLP are two powerful tools that can be used to process and analyze text. IE is the process of identifying and classifying named entities in text. At the same time, NLP is a field of computer science that deals with the interaction between computers and human (natural) languages.

The paper is structured as follows: Section 1 introduces the topic of text mining for healthcare. Section 2 provides a detailed analysis of the proposed healthcare solution and data collection procedures. Section 3 discusses the previous research on this topic and the gaps and limitations of this research. Section 4 of the paper discusses the in-depth healthcare analysis. It covers clinical decision support, healthcare administration, and public healthcare. Section 5 discusses the findings, challenges, limitations, and future work. Section 6 concludes the paper.

## 2. Material and Methods

In order to help scientific authors report a wide range of systematic reviews and meta-analyses, PRISMA (Preferred Reporting Items for Systematic Reviews and Meta-Analyses) is an evidence-based minimum set of items that are designed to help them assess the benefits and harms of a variety of healthcare interventions [54]. Database searches yielded 3128 records, and gray literature searches yielded 13. After removing duplicate articles and reviewing abstracts, the full text of 409 articles is reviewed. A flow chart illustrating the PRISMA process (identification, screening, and inclusion) [55] is shown in Figure 1.

In the identification process, each database is searched individually, including all search terms and truncations. Each database’s final number of records or articles is determined once all search terms are combined and all applicable limits are applied. Additionally, many researchers add notations in the box to indicate how many results there were for each database search, which could vary from PubMed (*n* = 1186) to healthcare (*n* = 84), or vice versa. The articles appearing more than once in search results are removed to avoid reviewing duplicate articles. A screening process is performed to determine whether the records are relevant to the research query, which is included in the full-text, according to certain criteria. To the right, note the number of articles that can find the full text that were unable to find it during text screening, and list any articles that cannot find the full text obtained in preparation for the full text screening (i.e., Screening Full-Text Records). Adding the record that is not retrieved to the number of reports sought for retrieval should give you this number. Review their full texts to determine whether these articles qualify for inclusion in the “Include and Exclude Report” systematic review. Enter the number of articles removed from the full-text screening process in the “Reports excluded” box, along with a description of the exclusions’ causes and the number of records removed for each reason. This study identified several shortcomings, such as incorrect settings, populations, interventions, and doses. To calculate the number of excluded articles, subtract the number of records excluded during the eligibility review of full-text articles from the total number of articles reviewed for eligibility. This method counts excluded articles once in the list, even if they meet multiple exclusion criteria. If necessary, combine the number with the literature search results in the box labeled “Studies included in the review.”

### 2.1. Datasets and Tools

Annotations of high quality are crucial to both the learning process and the evaluation process of a model. The following summarizes widely used datasets in unilingual tools (English) in NER. Tags are created using a corpus of documents containing annotations for multiple entities [56]. Data sources and the number of tags or entities are widely used for some datasets, as NER tasks are annotated, mainly, on news articles with a few entity types. The tag types used in MUC-2 to MUC-7 become significantly similar [57]. Along with these domain-specific datasets, a number of text-based datasets were developed with OntoNote information, as shown in Table 1.

Several recent NER studies used the English-based Conference on Natural Language Learning 2003 (CoNLL03) and OntoNotes Release 5.0 (OntoNotes05 (https://catalog.ldc.upenn.edu/LDC2013T19 (accessed on 8 January 2023)) datasets. However, with OntoNotes, 2005 Wikipedia, a huge corpus containing multiple genres is annotated with shallow semantics and structural information. A total of 18 tag types for entities are annotated in the texts. The English dataset contains a considerable percentage of news with annotations in three entity types (organization, place, and location). Table 1 summarizes the popular ones for multilingual (non-English) NER by academia and industry. Reuters News annotations in two languages are included in CoNLL03: German and Dutch. The main content contains five terms: clinical manifestation, syndrome, disease, treatment law, and herbs. Other side content information for Traditional Chinese Medicine (TCM) clinical treatment records includes a patient’s name and age. The datasets are generally very complex, large, hierarchical, heterogeneous, and differ in features [58,59]. Data preprocessing and transformation are required even before mining and discovery are applied.

Formal techniques can be used to check the strict correctness of AI-based solutions, as AI terms broadly include closely related areas (such as machine learning) [60,61]. Mathematical proofs, simulations, and testing are some methods that can be used. However, mathematical proofs provide the highest level of assurance, while simulations and testing are less time-consuming but do not provide the same level of assurance. In addition to formal techniques, a number of other things can be done to check the correctness of AI-based solutions, especially when it comes to gathering and processing data. These include:Data provenance: This involves tracking the origin of data and how it has been processed. This can help to identify any errors or biases in the data.Data quality: This involves checking data quality to ensure it is accurate, complete, and consistent.Data auditing: This involves reviewing the data to identify errors or biases.Data visualization: This involves using graphs, charts, and other visuals to represent data in a way that makes it easy to understand. This can help to identify any errors or biases in the data.Expert review: This involves having an expert review the AI solution to ensure it is correct.

These steps make it possible to check the correctness of AI-based solutions, especially when gathering and processing data.

### 2.2. Deep Learning Role in bNER

DL identifies the data representations at multiple levels by combining multiple processing layers with learning representations of data [62]. Traditional machine-learning techniques had limitations in processing raw natural data. In the forward pass, a weighted quantity of the inputs from the earlier layer is computed (via a non-linear function), and the result is passed through a nonlinear function. This study provided a comprehensive overview of a DL-based recommendation, which is analyzed within four dimensions: utilizing a system, remedies for challenges, awareness, purposeful properties, and prevalence recommendation system domains [63,64]. Machines are used for representation learning, i.e., to learn how to identify or classify data from raw data by feeding them raw data and automatically discovering the necessary representations. In DL, representations are addressed at multiple levels, each of which is derived by combining simple but non-linear modules to transform the original representation into an advanced abstract representation. DL provides representation learning and semantic composition capabilities as a function of both vector representation and neural processing. It allows raw data to be fed into a machine that automatically discovers latent representations, as well as processing that is required for detection or classification. In DL-based models, non-linear activation functions are used to learn complex and intricate features by comparing them to linear models.

In addition, NER features are designed significantly more efficiently using DL. It requires considerable engineering expertise and domain knowledge to use feature-based approaches. Meanwhile, DL-based models automatically extract useful representations from raw data. In addition, end-to-end bNER models undergo end-to-end gradient descent (GD) training [11]. In order to design potentially complex bNER systems, it takes advantage of this property. The existing taxonomy describes encoders at the character and word levels, as well as tag decoders [8,65,66]. We argue that the encoder at the word level in a DL-based NER model uses inaccurate word-level information. The EHR word-level and character-level representations are used to gather raw features and context dependencies for tag encoders and decoders. This study summarizes recent improvements in bNER, presenting the general architecture for the treatment and the journey of cancer patients alongside healthcare professionals, as shown in Figure 2.

Healthcare professionals detect the patients’ symptoms through diagnosis, which is treated through chemotherapy (in cancer) and other scientific procedures [67]. Each patient’s data is recorded through electronic information storage devices for future usage, treatments, and research. In addition to embedding at the word and character levels, distributed representations also incorporate features such as POS tags and gazetteers that are proven to be effective in feature-based representations. In context encoding, Convolutional neural networks (CNNs) [68], recurrent neural networks (RNN), or other networks are used to capture context dependencies. The decoder predicts tags in the input sequence based on the input sequence data. It is predicted that each token will have a tag of BIOES, indicating the B-(begin), I-(inside), O-(outside), E-(end), and S-(singleton) of the named entity [69]. In addition to detecting entity boundaries, the tag decoder is trained to classify and detect text spans to entity types by detecting entity boundaries. Note that the tag sequence or tag notations are not controlled during the training process when implementing the transformers-based (RoBERTa) model [70]. Conditional random fields (CRF) is used to control the sequence of each tag with tokens. An analysis of word shift plots is performed to determine what creates the dip in note scores after 21 days, as shown in Figure 3.

The ‘words’ associated with lung cancer treatment, such as ‘mouth’, ‘chemotherapy’, ‘dose’, ‘lung’, ‘treatment’, and ‘cancer’ perform to drive the score down. While more broad words, such as ‘support’, ‘care’, ‘independent’, ‘patient’, and ‘activity’, associated with patient care, inspire the note scores on peak days [18].

### 2.3. Deep Learning-Based Healthcare System

DL is an emerging technique of artificial intelligence used in several applications, including healthcare monitoring [71], speech recognition [72], computer vision [73], portfolio optimization [74], NLP [75], social network analysis [76], visual data processing, etc. Data analysis is becoming increasingly important in creating patterns and assisting in decision-making today. Moreover, this state-of-the-art method is capable of improving learning execution, expanding the scope of the research, and simplifying the measurement process. Health propaganda detection significantly deviates from other clarifications and, thereby, arouses doubt that a different mechanism in treatment causes it. Several detection techniques have been developed to solve similar problems. DL is one of the most effective methods for detecting misinformation within social networks and electronic data. In this case, DL techniques are applied to the same type of misinformation and the healthcare propaganda problem. Moreover, these DL techniques are dependent on a variety of data features, and they detect propaganda automatically.

DL algorithms classify tweet data into non-propaganda and propaganda classes [77]. Together with CNNs, long-short term memory (LSTM) [4], and Transformers-based Bidirectional Encoder Representations from Transformers (BERT) [78] language models in bNER, numerous researchers are driving the development of DL networks. Several studies have demonstrated that deep neural networks have improved NER in the biomedical field, such as BioNER [79], PubTator [80], Hunner. [81], GeneView [82], GENIA [83], and BioCreative [84]. Furthermore, word2vec models are compared [85], including FastText models [86], ELMo models [87], and a well-tuned BERT model [88], on a closed biomedical domain corpus to see how they contribute to the deep neural network (DNN). Persuasion is categorized as disinformation, which defines propaganda as a deliberate and systematic process to influence a person’s opinions, thoughts, and behavior in order to attain the intended aim [35]. Beside healthcare, propaganda plays an important role in politics, as it is used, primarily, to gain people’s faith in a person or a group. Governments and other organizations generally use propaganda for fast influence, which is either true or false [89]. Propaganda comes in several forms: text, images, and videos. In conclusion, researchers previously found that propaganda is related to healthcare (COVID-19), political discussions, and sectarianism.

Approximately 60% of the population is affected by Diabetes Mellitus, which is a disease with a high mortality rate. This study uses a multi-expert system ensemble model to diagnose diabetes type II, according to expert systems [90], to analyze various DL techniques for binary classification regarding illness, i.e., to diagnose whether a patient is suffering from disease or not [91]. Public authorities and governments urge consumers to verify the authenticity of circulating news on social media presenting misinformation about health, resulting in anxiety and panic among patients diagnosed with COVID-19 [92]. A data-mining analysis of information distribution on major social media platforms exposed COVID-19 rumors on a large scale by using information diffusion on COVID-19 [93]. Millions of people are affected by the COVID-19 spread worldwide, mostly in Europe, the United Kingdom, Spain, the United States, and the Asian subcontinent, and COVID-19 is the subject of extensive research for developing drugs to treat it. In order to cure this deadly virus, misinformation about how to treat it is spread extensively. Some of this misinformation claims that drinking alcohol or drinking cow urine will cure the virus. However, none of these claims is medically proven to exist. The World Health Organization suggested that the public take precautions against COVID-19 and has expressed concern about public health. Many hashtags are used to spread information about COVID-19 through social networks, such as Facebook and Twitter. Furthermore, the next section is an overview on the development of DL-based sentiment analysis.

### 2.4. Development of DL Sentiment Analyzer

To create a labeled data set where our machine learning sentiment analyzer can analyze the sentiment of a set of tweets, a manual coding approach is used, as with the topic classification. The reliability of the intercoder is measured by selecting a subsample of 300 Facebook reviews at random. The neutral or unidentified reviews are removed since we prefer binary sentiment classification. A stratified sampling technique is used to select training data for sentiment analysis, in which 80% of reviews within a class are selected for training. In order to train machine learning models, the Python packages (nltk, spacy, and scikit-learn) are utilized, which employ three basic classifiers: Naïve Bayes (NB), support vector machines (SVM), and logistic regression (LR). Several techniques are applied in this study to develop a sentiment analyzer [94,95,96,97,98], which analyzes patient online reviews to classify emerging and fading themes and sentiment trends during the early stage of the COVID-19 outbreak [99].

#### 2.4.1. Multi-Task Healthcare Analysis

The multi-task learning method uses the domain knowledge contained in training to inductively bias the training signals in an enhanced simplification by using domain information. The shared representation system learns tasks in parallel, which helps other tasks, in better ways, in parallel [100,101,102]. The multi-task algorithm can find internal representations used for various tasks, which learn regularities in a specific language and achieve POS, Chunk, NER, and SRL together. Lin et al. [103] combined transfer modules using two sharing parameters. A multilingual architecture for multi-task low-resource conditions effectively transfers dissimilar types of knowledge (auxiliary) to advance the main model. It is possible to use a multi-task learning framework in addition to NER, along with other tasks, to extract entities and relationships jointly [104]. bNER has developed four steps (data collection, annotation, model training, and evaluation) as two associated subtasks involving biomedical records and entity segmentation corpus likelihood [8,105,106,107].

Data collection: The first step is collecting a biomedical text corpus. This corpus should be large and diverse, and it should include a variety of biomedical entities. Subtasks are:Corpus cleaning: The corpus may need to be cleaned before it can be annotated. This may involve removing stop words, punctuation, and other noise from the corpus.Feature extraction: Features can be extracted from the corpus to help the machine learning model learn to recognize biomedical entities. These features can be lexical, syntactic, or semantic in nature.Annotation: The next step is to annotate the corpus with biomedical entities. This can be done manually or automatically. Subtasks are:Entity segmentation: The corpus is first segmented into entities. This can be done using a variety of techniques, such as rule-based systems or machine-learning models.Entity labeling: Once the entities are segmented, they are labeled with the appropriate biomedical entity type. This can also be done using a variety of techniques, such as rule-based systems or machine-learning models.

Model training: Once the corpus is annotated, a machine-learning model can be trained to recognize biomedical entities. This model can be trained using a variety of machine-learning techniques, such as support vector machines or neural networks. Subtasks are:Model selection: The first step is to select a machine-learning model. This can be done by considering the corpus size, the entities’ complexity, and the model’s desired performance.Model training: The model is then trained on the annotated corpus. This can be done using various techniques, such as batch training or online training.Model evaluation: A held-out test set evaluates the model’s performance. This ensures that the model is not overfitting the training data.

Evaluation: The final step is to evaluate the performance of the model. This can be done using a held-out test set or a cross-validation approach. Subtasks are:Precision: Precision is the fraction of entities that the model correctly identifies.Recall: Recall is the fraction of entities that are actually in the corpus and that are correctly identified by the model.F1-score: The F1-score is a weighted harmonic mean of precision and recall.

As NER on each dataset is considered a separate task in the biomedical domain, multi-task tuning is necessary to account for the differences between datasets [108,109]. In this case, the characters and words in the different datasets are assumed to be the same. As a result of this multi-task learning, the data is analyzed more efficiently, and more generalized representations are learned with the models.

#### 2.4.2. Healthcare Mining from EHR

Researchers in the mining community are giving considerable attention to healthcare. Healthcare methods are “a series of activities aimed at diagnosing, treating and preventing diseases and improving a patient’s health” [110]. Medical treatment and healthcare organization processes can be discovered and analyzed with the help of process mining [111]. Analyzing the behavior of processes between healthcare organizations is another application of process mining. In Elhaj et al. [112], supervised machine learning models are compared to determine which model is better at evaluating patient triage outcomes. In addition, Pandya et al.’s [113] InfusedHeart Framework outperformed RNNs, LSTMs, Bi-LSTMs, CNNs, K-means Clustering, and SVM-based classification approaches to extract more information from the recorded heartbeat sound signals.

#### 2.4.3. Healthcare Context in AI

Medical AI research presents unique challenges compared to research in other technical fields. A significant problem in medical diagnostics (e.g., the exact relationship between diseases and their causes) is the lack of such computable models. Even when it comes to the same clinical case, a doctor’s response may differ greatly from the response of another doctor. Consequently, it would be very challenging to train AI-based tools with biased responses, which are contaminated with discrete biases from clinicians who are ignorant of the truth of the reality in which they operate. The researchers are presented with the grand challenges of generating human-centered AI technologies that are ethically fair and enhance the human condition [114].

The healthcare industry is full of medical challenges with various characteristics, making AI research difficult. An AI model for cancer may not be able to be trained using the mathematical model developed for cardiovascular applications, for example, because the process is not generalized. Consequently, AI models tailored to cardiovascular applications cannot be trained. Moreover, the results may be inaccurate if underserved individuals and communities with infirmities are not considered during the AI system’s development. Depending on the application, specialized AI processes might be necessary. Several factors influence this, including healthcare decisions, target populations, data variability, and the quantity of helpful information contained within the data. Expert opinions are at the bottom of the hierarchy, followed by randomized controlled trials, systematic reviews, and meta-analyses, as shown in Table 2.

In recent years, expert opinions are outdated decision-making methods that have become obsolete. There is the possibility that an expert opinion may be unfitting, and often, there is disagreement among experts. Therefore, it occupies a low position in the hierarchy due to its low ranking. In clinical medicine, a series of cases refers to patients who share specific clinical, pathophysiological, or surgical characteristics used to identify particular disease features, treatments or diagnoses. According to the literature, high-quality evidence is needed for clinical practice, which is achieved traditionally with randomized controlled trials (RCT) [127]. Therefore, this is the decisive preview and exercise for patients. New drugs undergo evaluation phases, starting with pre-clinical, usually leading to the gold standard, which tests a larger population.

### 2.5. Tag Encoder-Decoder Architecture

The CRF is a statistical tool that evaluates the probability of a given observation sequence based on a global set of variables, and it is widely used in feature-based supervised learning approaches. CRF is used as a tag decoder in many DL models of neural networks, such as LSTMs and CNNs, which are built on top of each other and take advantage of the special properties of the CRF [128]. As shown in Table 3, the CRF tag decoder is the most commonly chosen tag decoder for CoNLL03 and OntoNotes5.0, and with the CRF tag decoder, state-of-the-art performance is obtained in these applications. As a result, CRFs, which are designed to fetch the information of a segment at the word level, cannot take full advantage of the information contained at the segment level because segment-level information is not encoded at the word level.

Among the many types of machine learning algorithms, the RNN is an increasingly popular one. In addition to solving or approximating increasingly difficult tasks, this RNN’s output is passed into a series of layers of RNNs; as explained below, short-term memory is a characteristic of RNNs. LSTM-Gates, coupled with a network, can generate long-term memory. This model updates the hidden unit at each time step, and the sequence length is not restricted. GRU-TN (gated recurrent unit tensor network) and LSTM-based models with bidirectional LSTM are proposed when dealing with a variety of sequence data types; they facilitate the generation of sequence data and improve the model’s flexibility [133,134]. Several RNN models, including BiRNN, divide the neurons into forward and backward computation with regard to their function [135,136]. Among the achievements of RNN are the recurrent gated units (GRU), along with the LSTM, which have shown great promise in modeling sequential data.

A biRNN effectively extracts input and future (positive and negative) information simultaneously for a specific timeframe, which is not the case with regular RNNs [137]. NER performance is enhanced with limited data by adding knowledge and data augmentation via a multi-task bidirectional RNN model combined with deep transfer learning [138,139]. The RNN encoder–decoder encodes a sequence of symbols into a static length vector representation of multi-RNN neural networks and symbols [140]. BiLSTM and CRF are applied as the basic RNN and DL models to encode and decode the context information [141,142,143,144,145,146,147,148]. According to Santos et al. [149], POS tagging is performed using DNN that learns character-level word representations and associates them with regular word representations. Recently, some studies [150] have integrated hypergraph representation, for nested entities, using linguistic features with designed LSTM-based neural networks [151]. Tan et al. [152] proposed the construction of a novel sequence-to-set neural network for nested NERs. A non-autoregressive decoder predicts the complete set of entities in one pass while capturing their interdependencies.

Pre-trained language models (PLMs) have become increasingly popular as NLP enters a new era. When a language model (LM) encounters a specific text set, it tries to reduce the amount of confusion that results [153]. A language model encoder is used for text production, and there are numerous different classifiers within a domain, so only one needs to be trained per domain [154]. PLMs are a very popular topic in NLP, and rapid development has led to the achievements of natural language today. Over time, as DL evolved, so did the number of model parameters. A significantly larger dataset is required to prevent overfitting, as shown in Figure 4.

## 3. In-Depth Healthcare Analytics

There are three application areas included in this review: clinical decision support, healthcare administration, and public health. It provides a detailed description of the operational definitions for these three areas.

### 3.1. Clinical Decision Support

An analysis of clinical decision support is aimed, mainly, at patients with diabetes, cancer, and cardiovascular disease, as well as patients in the internal care unit (ICU). Data-mining algorithms developed by some studies are reviewed in this article. Clinical decision-making papers are organized by major disease categories, with topics investigated and data sources used, as shown in Table 4.

### 3.2. Healthcare Administration

The majority of articles reviewed used data mining for administrative purposes in healthcare. Data mining is applied to various areas, including data warehousing and cloud computing, quality improvement, cost reduction, resource utilization, patient management, etc. Table 5 shows the number of articles, their focus areas, associated problems, and data sources.

### 3.3. Public Healthcare

Data mining is increasingly used to inform public healthcare decisions. Using data mining, researchers developed automated analytics tools for non-experts, identified resource utilization and patient satisfaction, and developed healthcare programs and emergency responses. It may be possible to develop a patient-centered, robust healthcare system if this effort is continued, as shown in Table 6.

## 4. Results

The most recent publications are listed first, so a summary of the 60 articles is in the qualitative analysis. There were 162 facilitators in the 60 articles and 97 barriers in relation to job satisfaction in long-term care facilities. The SVM model for binary relevance, as well as the classifier chain, had the highest F1-score (0.58) when examining the Tangible dimension. According to the F1-score for reliability and assurance, LR with binary relevance has the highest score, while NB with the label power set receives the highest score for responsiveness (0.63), and LR with the label power set receives the highest score for empathy (0.88). A consistently high F1 score is achieved only by SVMs with classifier chains in all service quality (SERVQUAL) dimensions (tangibility, reliability, responsiveness, assurance, and empathy). The five-fold cross-validation performance metrics are listed in Table 7.

In spite of the lower average accuracy compared to prior supervised machine learning studies, we also observed high predictive accuracy across SERVQUAL dimensions and a high F1 score. There was a 0.63–0.74 accuracy range for tangible dimension, a 0.65–0.71 accuracy range for reliability, a 0.53–0.71 accuracy range for responsiveness, a 0.57–0.69 accuracy range for assurance, and a 0.71–0.78 accuracy range for empathy. A range of F1-scores is found, for tangible dimensions, ranging from 0.38 to 0.62; for dependability dimensions, it ranges from 0.76 to 0.81; for responsiveness, it ranges from 0.40 to 0.65; for assurance, it ranges from 0.64 to 0.70; for empathy, it ranges from 0.82 to 0.87.

The prediction performance using supervised machine learning is summarized in Table 8. The best-performing multi-label classifier and classification model, prediction performance ratings, ranged from 0.13 to 0.25, with an F1 value of 0.68 to 0.75, and the models correctly classified the reviews. Based on the SVM model and classifier chain method, overall, the SVM model with the F1-score of 0.75 has the highest accuracy (0.21). However, for the topic classification model, hamming loss is more significant because it measures how many class labels are mispredicted.

As compared to all other models, the SVM model with a classifier chain is the lowest hamming loss (0.27). After SVM with the classifier chain, SVM with binary relevance is second, and five-fold cross-validation is used to evaluate all models.

## 5. Discussion

This review is focused on the unique role of AI in the treatment decisions of healthcare treatment (cancer). 

### 5.1. Summary of Finding

Extracting information from medical literature is a crucial aspect of biomedical informatics, which grows exponentially. bNER architectures are designed with DL-BiLSTM-CRF, using 1104 entities and 67,799 tokens for each annotator tag, as well as fine-grained biomedical corpus [169]. Specifically, 13 entity types are used in the language expressions in clinical records, revealing some interesting points from the corpus analysis. It is challenging to uncover the rules governing clinical records’ construction without analyzing the data. This analysis is used to identify the rules of healthcare and biomedical clinical expressions (entities) to generate questions for future research and summarize, as shown in Table 9.

### 5.2. Challenges of Text-Mining in Healthcare

Data mining has shown some promise in assisting healthcare researchers in modeling the healthcare system and improving healthcare delivery using its predictive techniques. However, mining healthcare data soon revealed many challenges attached to the validity of healthcare data and the limitations of predictive modeling, leading to the failure of data mining projects. Data-mining techniques and methods are gaining interest again, as the Big Data movement has gained momentum over the past few years.

DL techniques have been shown to be effective in detecting misinformation and healthcare propaganda [171]. However, a number of challenges must be addressed in order to improve the accuracy and reliability of these techniques. One challenge is the need for large and high-quality datasets. DL models require a large amount of data to train on, and this data must be of high quality in order to produce accurate results. However, it can be difficult to obtain large and high-quality datasets of misinformation and healthcare propaganda. Another challenge is the need to account for the diversity of misinformation and healthcare propaganda. Misinformation and healthcare propaganda can take many different forms, and it can be difficult to develop a single DL model that can effectively detect all forms of misinformation and healthcare propaganda. Finally, DL models can be computationally expensive to train and deploy. This can be a challenge for organizations that do not have the resources to invest in DL technology. Despite these disease challenges, DL techniques have the potential to be a powerful tool for detecting misinformation and healthcare propaganda. With continued research and development, DL models have improved to the point where they can be used effectively to protect people from the harmful effects of misinformation and healthcare propaganda.

In addition, to handle language ambiguity, we consider the annotations’ quality and consistency. Historical diagnoses are seeds that are incorporated into a patient’s comorbidity network as the patient’s comorbidities are expanded, and the disease risk is explored iteratively. The process of remote diagnosis is challenging since it is difficult to detect relevant data when observed remotely [14]. Patients may have different diseases based on the same historical diagnosis because they are different. In view of the sensitive nature of the data, it is impossible to differentiate between patients with similar historical diagnoses. An approach that uses data-driven approaches, based on the characteristics of their EHRs, is applied to predict disease, readmission times, and diagnoses of patients. EHR data is not utilized fully by most existing predictive models because some supervised events lack labels. Existing methods provide generic interpretability while being complex at the same time [39]. Data integration with multi-omics EHR data with low noise is challenging because it contributes to overfitting, poor generalization performance, and underfitting [41].

The clinical observations recorded in data mining are sparse, irregular, and highly dimensional, which creates significant challenges in estimating downstream clinical outcomes [42]. DL-based bNER systems are supervised with significant annotated data in training. Annotating data is an expensive and time-consuming process despite technological advances [56]. Annotation schemes commonly handle nested entities, fine-grained entities, and entities assigned multiple types. For many languages and domains with limited resources and skills, it is a big challenge to perform annotation tasks in a resource-poor environment since domain experts are necessary to accomplish these tasks.

Secondary data analysis is essential to ensure data quality and preprocess clinical data [59]. The treatments and diagnoses of diseases contain the needed information (for example, symptoms, diagnoses, prescriptions, and laboratory results). Since clinical data from multiple sources has different terms, big data management with lexicon sources is essential. These terms allow clinical data to be integrated, uniformly arranged, analyzed, and retrieved quickly. Developing a unified health information model is necessary for the efficient data analysis of large-scale clinical data. In addition, controlled vocabulary and terminology systems are used to enhance clinical data functionality and usability. Electronic medical records are used as clinical record structures in bNER. High data quality, preprocessing, and managing large-scale data, including multi-relational analyses based on domain knowledge, are key challenges.

### 5.3. Implications

This study provides useful managerial implications for healthcare data scientists, practitioners, and data analysts. Big Data has emerged recently, as biomedical data has grown explosively, and medical technologies and computers have advanced rapidly. Among all of the research articles presented in this study, one specific feature stands out: It discusses how clinical decision-making and ML approaches are used in a wide range of applications in healthcare. They include bNER using EHR data for treatment prediction, benefit package design based on healthcare needs, and research on the effects of health scheme design on healthcare utilization.

This section examines healthcare and sentiment analysis in bNER areas that have effectively used DL tools to create models to solve specific tasks using EHRs. Hence, healthcare data scientists and analysts can take advantage of the present study and the research articles in the analysis to gain a comprehensive and synthesized view of how this gap can be filled, with successful case studies, across all healthcare problems and all the issues involved in each healthcare treatment prediction.

Applying treatment prediction methodology, in practice, requires a thoughtful implementation study with public healthcare in the lead. Data science also needs a high-quality, up-to-date technical infrastructure.

### 5.4. Limitations and Future Work

This study discusses several limitations and research directions for future healthcare and text-mining researchers. First, this study did not analyze post-COVID-19 healthcare literature; therefore, future researchers may collect more data after the pandemic, as text-mining methods are more efficient with larger datasets. A possible future work to address this limitation could be to collect and analyze post-COVID-19 healthcare literature using a text-mining approach. This could involve identifying relevant healthcare topics in post-COVID-19 literature and quantifying the frequency and sentiment of those topics. In addition, NLP techniques were used to identify relationships between topics within the literature [99,172,173]. This could provide insights into how healthcare topics have changed after the COVID-19 pandemic.

Second, this study employed nested named entities, which can be difficult to identify and extract because they are not explicitly labeled in the text. Hence, nested entities pose a limitation to current NER tasks in the healthcare domain. To address this limitation, future research should focus on the development of improved annotation schemes to incorporate nested entities, more robust models to detect nested entities, and experimentation with different feature and model architectures. Moreover, it should be investigated how to better detect nested entities by including contextual information in the models.

Third, in this study, we expect to pay attention to bNER as modeling languages advance and real-world applications increase. bNER, on the other hand, is applied before downstream processing. Therefore, various clinical datasets with disparate characteristics of cancer patients permit further investigation to find the behavior of imputation in healthcare studies [42]. The sensitivity of the data makes it impossible to access other information, such as medical diagnoses, except for personal information.

Fourth, the model in this study cannot differentiate between patients with the same diagnosis. It uses co-occurring frequency as a form of external knowledge, but disease relationships in the EHR are not limited to this type of knowledge [14,39]. The length of the time series is also constrained, which limits the model’s ability to learn complex patterns. The study proposes a future direction of research to address these limitations by developing a model that can differentiate between patients with the same diagnosis, using additional forms of external knowledge and removing the length constraint on time series. As a result, EHR predictions would be more accurate and comprehensive [41].

Fifth, this study proposes a bNER system for EHRs in English. However, the system is limited to English text. To address this limitation, future work could develop a multilingual bNER system. This could be done by training a new bNER system on a dataset of biomedical text in multiple languages. Alternatively, future work could develop a bNER system that can automatically translate biomedical text into a single language before performing NER. This would allow the system to handle multilingual text without having to be trained on a dataset of biomedical text in each language.

## 6. Conclusions

A biomedical informatics system analyzes and identifies bNER and healthcare propaganda, as biomedical data moves from cellular to electronic form. Information Technology and computational techniques are needed to uncover new information about researchers and patients in the biomedical field. EHRs contain basic and relevant medical information in their text, identifying medical entities with special meanings, such as people, places, and organizations. The expansion of EHR and clinical data has been unprecedented, as well as the spread of misinformation, fakes, and propaganda through social media and the free-text internet. This study aims to review recent studies on DL-based bNER and health propaganda solutions to help potential researchers build a comprehensive understanding of this field. Additionally, we present some preliminary findings regarding classifying healthcare tasks, traditional approaches for data mining, basic concepts, and evaluation metrics in DL.

## Figures and Tables

**Figure 1 healthcare-11-01268-f001:**
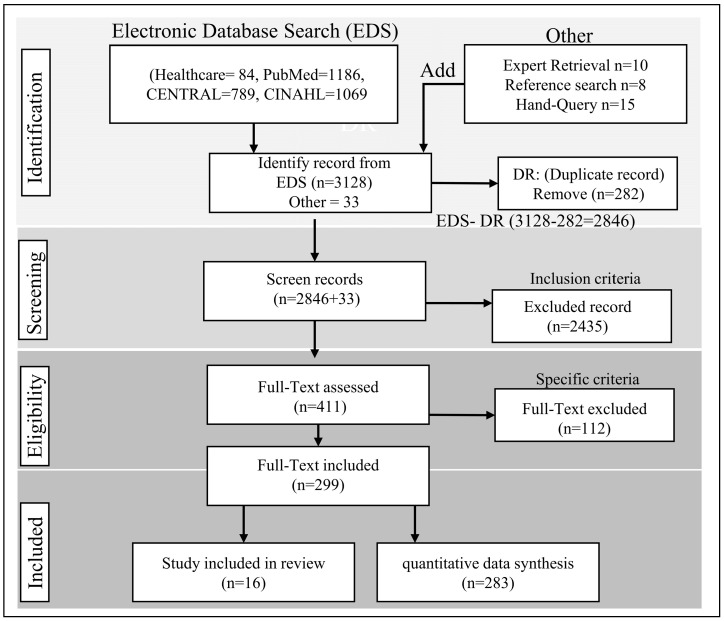
PRISMA flow diagram illustrating the steps involved in conducting a systematic literature review.

**Figure 2 healthcare-11-01268-f002:**
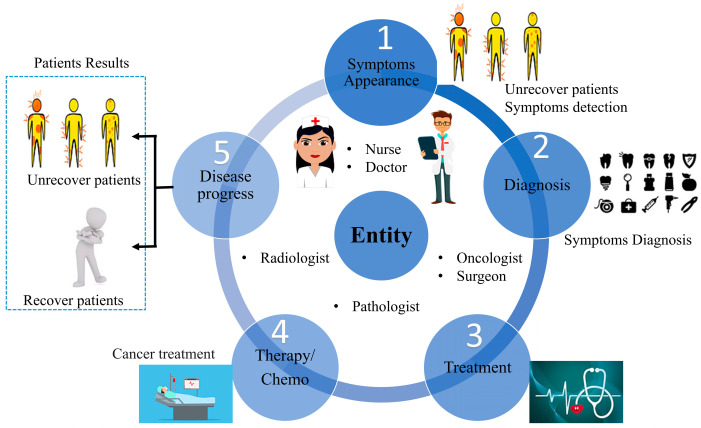
A general pipeline for the treatment of cancer patients along their journey through the healthcare system.

**Figure 3 healthcare-11-01268-f003:**
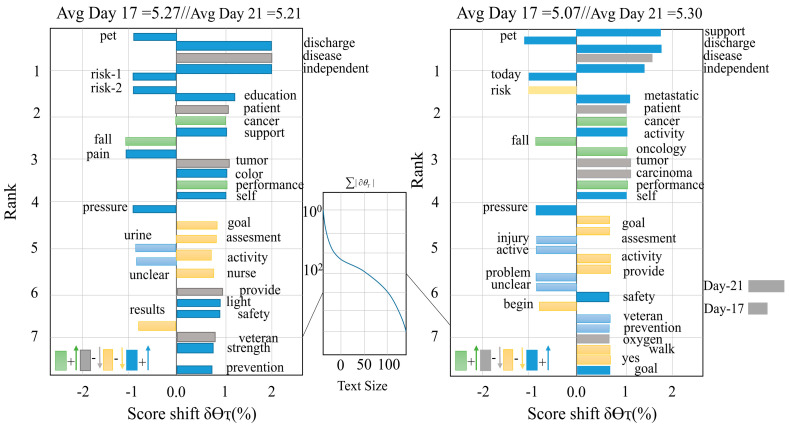
A visual representation of the decrease in positive sentiment between Day 19 and Day 21 of treatment.

**Figure 4 healthcare-11-01268-f004:**
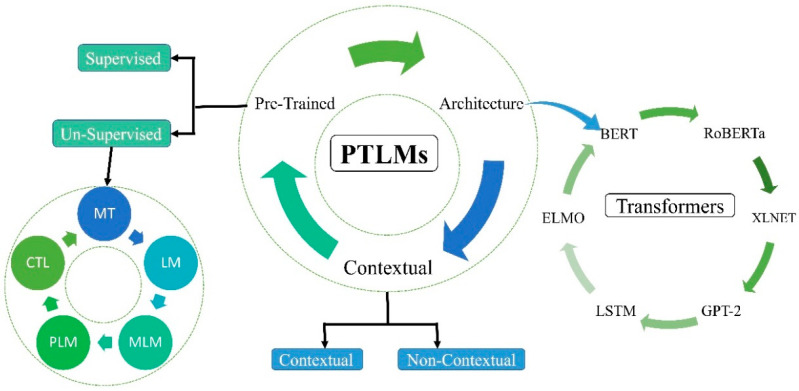
Pre-trained language models (PLMs) with typical examples.

**Table 1 healthcare-11-01268-t001:** Uniform Resource Locators (URL) for all monolingual NER datasets (English).

Corpus	Date	Text-Source	Tag	Data URL ^1^
MUC-2, MUC-3, MUC-4,	1995	MUC Data Sets		https://www-nlpir.nist.gov/related_projects/ (accessed on 5 January 2023)
MUC-6	1995	Wall Street Journal	7	https://catalog.ldc.upenn.edu/LDC2003T13 (accessed on 5 January 2023)
MUC-6 Plus	1995	Additional news	7	https://catalog.ldc.upenn.edu/LDC96T10 (accessed on 5 January 2023)
MUC-7	1997	New York Times news	7	https://catalog.ldc.upenn.edu/LDC2001T02 (accessed on 5 January 2023)
CoNLL03	2003	Reuters news	4	https://www.clips.uantwerpen.be/conll2003/ner/ (accessed on 5 January 2023)
ACE 2003	2003	Wikipedia	7	https://www-nlpir.nist.gov/ (accessed on 5 January 2023)
ACE 2005	2003	Wikipedia	7	https://catalog.ldc.upenn.edu/LDC2004T09 (accessed on 5 January 2023)
OntoNotes	2005	Wikipedia	18	https://catalog.ldc.upenn.edu/LDC2013T19 (accessed on 5 January 2023)
BTC	2016	Twitter	-	http://www.i2b2.org (accessed on 5 January 2023)
i2b2-2006	2007	Medical	-	https://www.i2b2.org/NLP/DataSets/Main.php (accessed on 5 January 2023)
CADEC	2015	Medical	-	http://data.csiro.au/ (accessed on 5 January 2023)
CONLL03	2003	News	4	https://github.com/synalp/NER/tree/master/corpus/CoNLL-2003 (accessed on 5 January 2023)
**BiLingual NER**				
UNER	2008	SSEAL	URDU	http://www.iiu.edu.pk/?page_id=5181 (accessed on 5 January 2023)
MK-PUCIT	2019	-	URDU	https://www.dropbox.com/sh/1ivw7ykm2tugg94/AAB9t5wnN7FynESpo7TjJW8la (accessed on 5 January 2023)
ACE-03	2003	Wikipedia	Ch/Ar	https://catalog.ldc.upenn.edu/LDC2004T09 (accessed on 5 January 2023)
ACE-04	2004	Wikipedia	Ch/Ar	https://catalog.ldc.upenn.edu/LDC2005T09 (accessed on 5 January 2023)
ACE-07	2007	Wikipedia	Ch/Ar	https://catalog.ldc.upenn.edu/LDC2006T06 (accessed on 5 January 2023)
QUAERO	-	French Medical	Fr	https://quaerofrenchmed.limsi.fr/ (accessed on 5 January 2023)
KIND	-	-	It	https://github.com/dhfbk/KIND (accessed on 5 January 2023)
hr500k	-	-	Cr	http://hdl.handle.net/11356/1183 (accessed on 5 January 2023)
CoNLL03	2003	German	Gr	https://www.clips.uantwerpen.be/conll2003/ner/ (accessed on 5 January 2023)
CoNLL03	2003	Dutch	Du	https://www.clips.uantwerpen.be/conll2002/ner/ (accessed on 5 January 2023)

^1^ URL is provided to assist the access of free available data.

**Table 2 healthcare-11-01268-t002:** AI in healthcare summary of recent and relevant articles.

Reference	Year	Method	Application	Field	Limitations	Key Features
[115]	2023	Neural network (NN)	Develops and implements a NN-based method for skin cancer prediction	Dermatology	This investigation failed to identify the optimal CNN architecture.	Image recognition
[116]	2023	DL systems	Comparing the performance of a DL against a retinal specialist vision-threatening all-cause referable	Ophthalmology	This study does not use: DLS-driven DR screening, a focus on probable validation, and real-world implementation.	Image recognition
[117]	2022	Neural network	Image segmentation review and six optimizations of the radiotherapy dose.	Oncology	Many-to-one mapping; intrinsic fitting error; patient-specific tissue heterogeneities	Image recognition
[118]	2022	Multi-channel LSTM	To signify a medical event that is strongly linked to the prediction task using EHRs	Medical event prediction	Fast knowledge and effectively managing the clinical result from intricate and heterogeneous EHRs	NLP
[119]	2022	LSTM-CNN	Twitter analysis and prediction of real-time COVID-19 data	Pathology	State-of-the-art results, implementing bidirectional LSTM	NLP
[120]	2022	ResNet-CNN-BiLSTM	To conclude, what kind of cyberbullying is being committed, and classify the individuals involved	Psychological care	The validity and reliability of the data, as well as biosensor and biomarker-based techniques, are affected by a number of factors, such as sample quality, instrument accuracy, and environmental conditions.	NLP
[121]	2021	NN	Diagnosing cardiovascular diseases using cardiac MRI images	Radiology	Lack of generalizability of the results, which ends with lesion classification, is not automated here during MRI analysis.	Image recognition
[122]	2019	NN-integrated gradients	Grading diabetic retinopathy more rapidly and precisely	Ophthalmology	Insufficient discussion or interpretation of the results. It is likely that the length of the grading process will affect the experience in some way.	Image recognition
[123]	2019	NN	To regulate which of the X-ray images showed a healthy chest	Radiology	Optimization of the probability threshold for classification of an Inception-ResNet-v2 model trained on ImageNet	Image recognition
[124]	2019	LR (Logistic regression)	Ultrasound inspections are used to establish the criteria for classifying breast cancer known as triple negative (TN).	Radiology	Lacking computerized segmentation of the masses, fewer people are involved in sampling bias.	Image recognition
[125]	2018	-	Multi-step processing, including deep-learning-based segmentation, revealed variability in the composition of tumor-immune		It offers a unique challenge when it comes to de-noising MIBI-TOF data due to two properties: Low intensity values, low abundant antigens, and signal is sparse and pixelated	Image recognition
[126]	2017	Neural network	To achieve dermatologist-level accuracy when diagnosing skin cancer	Dermatology	CNNs require large amounts of data to train. Collecting large amount of labeled skin cancer images is difficult and time-consuming.	Image recognition

**Table 3 healthcare-11-01268-t003:** CRF/LSTM is the state-of-the-art performance encoder/decoder for CoNLL03 and OntoNotes5.0.

Work	Input Representation		Encoder/Decoder ^1^	F1-Score
	Word	Character		
[129,130]	Google Word2vec	LSTM	LSTM/CRF	0.8409
[131]	Globel Vector	LSTM	LSTM/CRF	0.9337
[132]	Twitter Word2vec	CNN	LSTM/CRF	0.4186
[88]	WordPiece	-	Transformer/Softmax	0.9208

^1^ Decoder is the most normally chosen tag.

**Table 4 healthcare-11-01268-t004:** The major disease categories and topics using clinical decision-making are arranged here, as well as the data sources for the papers.

Author	Symptom ^1^	Data Source	Topic Relation
Tsipouras et al. [155]	Coronary disease	Cardiology department University Hospital Greece	Coronary heart disease treatment
Rakha et al. [156]	Cancer	Breast cancer data set, Wisconsin	Breast cancer classification patients with innovative algorithm
Tapak et al. [157]	Diabetes	Iranian national non-communicable diseases risk factors surveillance	Evaluation of diabetes classification accuracy algorithms
Iqbal et al. [158]	Cancer	Taiwan National Health Insurance Database	Visualization tool for cancer
Agrawal et al. [159]	Lung cancer	SEER Program of the National Cancer Institute, USA	Lung cancer survival prediction
Yeh et al. [160]	Other	Hemodialysis center in Taiwan	Hospitalization prediction of Hemodialysis patients

^1^ Data mining is applied to identify rare forms of diabetes, coronary heart Disease, and cancer.

**Table 5 healthcare-11-01268-t005:** Data sources and problems in healthcare administration with regard to the analysis of problems.

Author	Research ^1^	Data Source	Limitation Analysis
Arvisais et al. [161]	cloud computing	Not specified	Reports from radiology repositories
Pur et al. [162]	Utilization, quality, and costs of healthcare	National Institute of Public Health; Health Care Institute, Celje; Slovenian	Framework and model for monitoring HC networks’ resource allocation
Koskela et al. [163]	Management of patients	Tampere Health Centre, Finland	Predicting persistent healthcare attendance by examining risk factors
Oliwa et al. [164]	Healthcare quality	Pathology company in Australia	Efficient pathology ordering organization

^1^ Keeping electronic patient records secure and cost-effective requires data warehousing and cloud computing.

**Table 6 healthcare-11-01268-t006:** Data sources and problems in healthcare administration with regards to the analysis of problems.

Author	Research	Data Source	Limitation Analysis
Nimmagada et al. [165]	Healthcare	World Health Organization	Programs designed to prevent health problems
Kostkova et al. [166]	Healthcare quality	National electronic Library of Infection	Public and professional information seeking behavior
Rathore et al. [167]	Healthcare data	Internet of Things and big data for real-time emergency response	UCI machine learning depository
Lavrač et al. [168]	Healthcare quality	Slovenian national Institute of Public Health	Innovative use of data mining

**Table 7 healthcare-11-01268-t007:** The five-fold cross-validation metrics for each SERVQUAL dimension in MLQC [97].

^1^ Metrics	Tangible			Reliability			Responsive			Assurance			Empathy		
	NB	SVM	LR	NB	SVM	LR	NB	SVM	LR	NB	SVM	LR	NB	SVM	LR
Binary relevance															
Acc	0.675	0.7	0.68	0.69	0.71	0.71	0.63	0.64	0.65	0.64	0.71	0.73	0.78	0.78	0.79
F1	0.39	0.58	0.47	0.81	0.82	0.81	0.48	0.55	0.54	0.68	0.72	0.73	0.87	0.87	0.88
P	0.76	0.69	0.67	0.68	0.76	0.71	0.66	0.61	0.67	0.6	0.71	0.71	0.78	0.8	0.79
R	0.27	0.51	0.36	0.99	0.88	0.97	0.39	0.51	0.46	0.79	0.73	0.75	1	0.8	0.99
Label powerset															
Acc	0.66	0.6	0.64	0.69	0.68	0.7	0.55	0.61	0.61	0.56	0.63	0.67	0.78	0.78	0.8
F1	0.53	0.52	0.48	0.81	0.79	0.81	0.63	0.61	0.62	0.67	0.68	0.67	0.87	0.87	0.88
P	0.61	0.61	0.57	0.69	0.72	0.71	0.5	0.55	0.55	0.52	0.59	0.56	0.78	0.81	0.8
R	0.49	0.4	0.43	0.99	0.88	0.92	0.87	0.68	0.73	0.94	0.81	0.82	0.9	0.97	0.9
Classifier chain															
Acc	0.67	0.71	0.68	0.69	0.73	0.71	0.63	0.65	0.64	0.65	0.63	0.71	0.78	0.79	0.79
F1	0.399	0.58	0.47	0.81	0.81	0.82	0.47	0.57	0.57	0.68	0.72	0.7	0.87	0.87	0.88
P	0.76	0.69	0.67	0.68	0.76	0.71	0.67	0.63	0.61	0.61	0.72	0.71	0.78	0.82	0.8
R	0.27	0.51	0.37	0.9	0.87	0.94	0.37	0.53	0.5	0.7	0.74	0.7	1	0.92	0.98
Rak-EL															
Acc	0.63	0.71	0.67	0.69	0.7	0.71	0.62	0.63	0.65	0.64	0.68	0.69	0.78	0.78	0.79
F1	0.27	0.58	0.49	0.81	0.8	0.81	0.52	0.55	0.56	0.65	0.68	0.69	0.87	0.8	0.88
P	0.68	0.7	0.65	0.69	0.73	0.73	0.65	0.59	0.64	0.63	0.66	0.67	0.78	0.87	0.8
R	0.17	0.49	0.39	0.9	0.9	0.93	0.5	0.52	0.52	0.71	0.71	0.72	1	0.95	0.98

^1^ The main evaluation metrics are Accuracy (Acc), Precision (P), Recall (R), and F1-score; Naïve Bayes (NB), support vector machines (SVM), and logistic regression (LR), as well as Random k-Label sets (Rak-EL) are used.

**Table 8 healthcare-11-01268-t008:** The performance classification of multi-label models is based on the five-fold cross-validation of the ML models [97].

Classifier		**Accuracy**	**F1-Score**	**Precision**	**Recall**
Binary Relevance	NB	14.71%	0.73	0.70	0.76
	SVM	21.1%	0.75	0.74	0.76
	LR	19.3%	0.75	0.73	0.77
Classifier Chain	NB	14.9%	0.74	0.63	0.89
	SVM	21.5%	0.73	0.67	0.79
	LR	19.1%	0.73	0.66	0.82
Label Powerset	NB	13.0%	0.73	0.70	0.75
	SVM	16.6%	0.75	0.75	0.76
	LR	15.8%	0.74	0.72	0.77
ML-KNN, BR-KNN ^1^	NB	18.0%	0.72	0.69	0.74
	SVM	14.0%	0.74	0.72	0.76
Rak-EL	NB	15.7%	0.74	0.72	0.76
	SVM	18.6%	0.71	0.69	0.73
	LR	18.0%	0.68	0.73	0.64

^1^ Multi-Label K-Nearest Neighbors algorithm (MLKNN); Binary Relevance K-Nearest Neighbors (Br KNN), Random k-Label sets (Rak-EL).

**Table 9 healthcare-11-01268-t009:** Summaries of the clinical expressions in healthcare and biomedicine.

Author	Year	Method	Study Population	Design	Summary	Objective
Shickel et al. [2]	2018	DL to clinical tasks	EHR Projects	Review	Data mining for clinical tasks based on EHRs. Applications include information extraction, representation learning, outcome prediction, and phenotyping.	Model interpretability, data heterogeneity, and the lack of universal benchmarks are among the limitations found in current research.
Festen et al. [13]	2021	Descriptive statistics	Two hundred fourteen patients with cancer of 70 year Recruited from a University Medical Center, Netherlands.	Quantitative	(1) Implementing nurse-driven geriatric assessment. (2) Evaluate the patient’s preferences regarding treatment outcomes. (3) Considered patient assessments in their decision-making.	The geriatric assessment of older cancer patients is incorporated into a novel care pathway. A multidisciplinary team (MDT) of oncogeriatric experts could modify treatment.
Quinn et al. [15]	2019	Content analysis.	patients 32 (17 stable, 15 unstable	Survey	Healthcare team confidence and connection are associated with stable HCT. In contrast to complete independence or autonomy, interdependence involves building connections with healthcare providers and eliciting their support.	Researchers determined that resilience constructs play a protective role in helping AYA with KT maintain stable HCT.
Song et al. [49]	2021	DL-based methods for BioNER and datasets	Biomedical text mining	Review	Algorithms used in deep learning can be categorized into four types: that which is based on one neural network, that which is based on multi-task learning, that which is based on transfer learning, and that which is based on hybrid models.	Dataset size and type determine the results of these algorithms when applied to BioNER in a wide variety of domains.
Bridge et al. [67]	2015	Face-to-face interviews. Framework Analysis.	Healthcare professionals (*n* = 22; nurse; *n* = 11 physician) Recruited NHS hospital in UK.	Qualitative	(1) Provide quality, timely, and accurate information to patients. (2) In the meeting, patients’ complex information is given attention. (3) Promoting patient autonomy. (4) Participate in multidisciplinary teams.	Identify factors shaping cancer treatment decisions for older people with complex needs, including health and social care.
Jones et al. [170]	2018	Semi-structured interview.	Thirty-five sets of patients and their decision partners, Recruited from Cancer Center, USA	Qualitative	(1) Making the discussion more thorough between patients and decision partners with the use of decision aids. (2) Facilitating patient-provider relationships via decision aids.	Patient and partner experiences with an interactive decision aid utilized to make informed, shared treatment choices for advanced prostate cancer.
Peng et al. [117]	2022	AI utilization in particle therapy	Image segmentation review and six optimization of the radiotherapy dose.	Oncology review	Research on AI-powered particle therapy based on a comprehensive literature review. Creating data, interpreting it, incorporating fundamental physical processes, and rigorously validating it are all as important as the actual AI model development.	The aim is to present six aspect included: dose calculation, treatment planning, image guidance, range and dose verification, adaptive replanning and quality assurance

## Data Availability

Not applicable.

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
