# Peer review of "A Review on Electronic Health Record Text-Mining for Biomedical Name Entity Recognition in Healthcare Domain"

_healthcare, 2023, doi:10.3390/healthcare11091268_

Round 1

Reviewer 1 Report

Summary: " This paper reviews the healthcare domain of bNER using deep learning (DL) techniques and artificial intelligence in clinical records for mining treatment prediction. bNER-based tools is categorized systematically and represent the distribution of input, context, and tag (encoder/decoder)."   Suggestions and Remarks:   1. The paper is well-written and well-structured. In addition, the topic considered is quite interesting.   2. Please add a paragraph that describes the structure of the paper at the end of the introduction.   3. The introduction is a bit long and may be split into two sections.   4. Line 179: "Many recent NER studies is used the English-based CoNLL03" ===> English mistake   5. A deep proofread is needed.   6. The authors are asked to add a short paragraph about how formal techniques can be used to check the correctness of AI-based solutions, especially when it comes to gathering and processing data.     7. For this purpose, the following references may be included: a. https://ieeexplore.ieee.org/document/9842406 b.  https://dl.acm.org/doi/abs/10.1145/3503914   8. Figure 3 is small and needs to be enlarged.   9. Table 2: It will be useful to insert a new column that indicates the limitations of each considered paper.   10. The authors are invited to identify the limitations of their work and propose some future work directions.

Reviewer 2 Report

A Review on Electronic Health Record Text-Mining for Biomedical Name Entity Recognition in Healthcare Domain

This paper reviews the healthcare domain of bNER using deep learning (DL) techniques and artificial intelligence in clinical records for mining treatment prediction. bNER-based tools is categorized systematically and represent the distribution of input, context, and tag (encoder/decoder). To conclude, we discuss the challenges facing bNER systems and future directions in the healthcare field.

Some of the suggestions needs to addressed

In introduction the authors can come up with the existing survey works on the similar topic, probably summary table.

Figure 2. There is a general pipeline for the treatment of cancer patients along their journey through the healthcare system is clear but needs to bit informative

DL techniques is applied to the same type of misinformation, and healthcare propaganda problem. Moreover, these DL techniques are dependent on a variety of data features and detect propaganda automatically. The authors should mention the challenges of DL

bNER is developed four steps as two associated subtasks involving biomedical records and entity segmentation corpus likelihood are there any sub steps involved please check and clarify

There should be connection between one section to the next section, the flow is bit harder.

The main contribution of the survey lies in challenges and future directions these are missing.

Add more challenges on the current trends and latest technologies.

The authors can refer the latest works: InfusedHeart: A novel knowledge-infused learning framework for diagnosis of cardiovascular events

Deep learning disease prediction model for use with intelligent robots

Reviewer 3 Report

Dear authors,

The work may be published, but the following improvements should be considered:

- Improve the quality of figures.

- General revision in the text, for a few corrections in the writing (For example e line 5, should be written Technology with a capital letter).

Reviewer 4 Report

As biomedical text-mining progresses from molecular to electronic levels, the system is designed for biomedical informatics, and it identifies biomedical name entity recognition (bNER) in healthcare systems. Information about biomedical researchers, health workers and patients is needed to discover novel knowledge using computational methods and information technology. bNER is contained the basic and pertinent medical information in data mining that identifies medical entities with special meanings like people, places, and organizations as predefined semantic types from electronic health records (EHR). EHR and clinical data is expanded unprecedentedly, and with social media, misinformation and propaganda is spread unprecedentedly. Early name entity (NE) systems is configured manually to include domain-specific features and rules, but they achieved successful implementation results.

The authors review the healthcare domain of bNER using deep learning (DL) techniques and artificial intelligence in clinical records for mining treatment prediction.

They categorized the bNER-based tools systematically and represent the distribution of input, context, and tag (encoder/decoder).

They also reported the challenges facing bNER systems and future directions in the healthcare field.

Interesting study.

I have some suggestions for the authors:

1.       The abstract must be rewritten better summarizing the sections.

2.       Describe fig. 2  in details. Follow this suggestion also for the other figures

3.       Improve the resolution in figure 3

4.       Insert the limitations of your study

Round 2

Reviewer 1 Report

The authors considered all my comments and suggestions. Good luck.

Author Response

We greatly appreciate getting the opportunity to revise our manuscript and your provision of valuable feedback on our work.

Reviewer 2 Report

Paper can be accepted 

Author Response

We are grateful for your interest in our work and for providing us with valued comments.